# A 16-Year Single-Center Series of Trachea Resections for Locally Advanced Thyroid Carcinoma

**DOI:** 10.3390/cancers16010163

**Published:** 2023-12-28

**Authors:** Julia I. Staubitz-Vernazza, Sina Schwind, Oana Lozan, Thomas J. Musholt

**Affiliations:** Section of Endocrine Surgery, Department of General, Visceral and Transplantation Surgery, University Medical Center, Johannes Gutenberg University Mainz, Langenbeckstraße 1, D-55131 Mainz, Germany; julia.staubitz-vernazza@unimedizin-mainz.de (J.I.S.-V.); sinafuhrmann@gmx.net (S.S.); oana.lozan@unimedizin-mainz.de (O.L.)

**Keywords:** trachea infiltration, thyroid carcinoma, trachea resection

## Abstract

**Simple Summary:**

Infiltration of the aerodigestive tract determines the prognosis in advanced thyroid carcinoma, ultimately leading to suffocation. To avoid this fatal outcome, surgical treatment strategies—within the framework of multidisciplinary patient care—are required. There are no established guidelines to rely on, and an individualized surgical treatment approach is required. The aim of this study was to present and critically analyze the stepwise escalatory surgical treatment strategy, which was applied over a 16-year time period at the University Medical Center Mainz, in respect to the resulting recurrence-free survival. The feasibility of stage-adapted tracheal resections is illustrated by the present study. Despite the risk of potentially life-threatening complications from the operations, prolonged recurrence-free and overall survival were observed in the institutional cohort of patients suffering from high-stage thyroid carcinoma.

**Abstract:**

(1) Background: Infiltration of the aerodigestive tract in advanced thyroid carcinoma determines the prognosis and quality of life. Different stages of tracheal tumor invasion require customization of the surgical concept. (2) Methods: In the period from January 2007 to January 2023, patients who underwent surgery for advanced thyroid carcinomas with trachea resections were included in a retrospective observational study. The surgical resection concepts and operation-associated complications were documented. The overall survival and post-resection survival were analyzed. (3) Results: From 2007 to 2023, at the single-center UMC Mainz, 33 patients (15 female and 18 male) underwent neck surgery with trachea resections for locally advanced thyroid carcinomas. Of these, 14 were treated with non-transmural (trachea shaving) and 19 transmural trachea resections (9 “window” resections, 6 near-circular resections, 3 sleeve resections and 1 total laryngectomy with extramucosal esophageal resection). The two-year postoperative survival rate was 82.0 percent. The two-year recurrence-free survival rate was 75.0 percent (mean follow-up period: 29.2 months). (4) Conclusions: Tracheal resections for locally advanced tumor infiltration are feasible as an element of highly individualized treatment concepts.

## 1. Introduction

In 2018, the incidence of thyroid carcinoma in Germany was 10.2 per 100,000 females and 4.7 per 100,000 males [1]. For both sexes, these carcinomas were primarily diagnosed at Union for International Cancer Control (UICC) stage I (88 and 75 percent, respectively) [1,2], which generally allows for curation after thyroidectomy (potentially complemented by central lymphadenectomy) and radioiodine therapy in the case of differentiated thyroid carcinoma. Only in 5 and 11 percent (females and males, respectively) advanced thyroid carcinomas (UICC III and IV) were present at the time of diagnosis [1,2]. Particularly for UICC stage IV, the relative 5-year survival was markedly reduced, 59 percent in females and 56 percent in males, whereas the overall relative 5-year survival for thyroid carcinoma in Germany was 95 percent and 91 percent, respectively [1,3]. For invasive thyroid carcinoma, the trachea was described as the most common site of aerodigestive tumor invasion (incidence of 35–60 percent among the patients with invasive tumor growth), followed by invasion into the esophagus (21 percent) and the larynx (12 percent) [4,5,6,7,8]. In comparison, 85–95% of cancers affecting the larynx originate from the entity of head and neck squamous cell carcinoma [9].

Risk factors for aerodigestive tract invasion include poorly differentiated histology, aggressive histologic growth patterns, advanced patient age and recurrent disease [6,10]. Because locally advanced disease was shown to have a significant impact on survival, the choice of the adequate resection technique for these carcinomas holds important value for patients suffering from this condition. Surgical resection techniques, adapted to the different stages of laryngotracheal tumor invasion by thyroid carcinoma, were first reported by Shin et al. in 1993 and further described by Dralle et al. and Brauckhoff et al. [11,12,13]. These include non-transmural tracheal resections (shaving), transmural trachea resections (window resection and sleeve resection) and cervical evisceration. The present article reports the resection strategies and clinical outcome of patients undergoing surgery for advanced thyroid carcinoma with trachea invasion at the single center of the University Medical Center Mainz in a 16-year time period.

## 2. Materials and Methods

From 2007 to 2023, all patients undergoing neck surgery including trachea resections at the Section of Endocrine Surgery of the single-center UMC Mainz were analyzed in a retrospective study. Patients with tumor entities other than thyroid carcinomas were excluded from the present analysis (Figure 1).

If an infiltration of the cervical–visceral axis was preoperatively known or suspected, panendoscopy was performed at the Clinic for Otorhinolaryngology of the UMC Mainz. Furthermore, vocal cord function was assessed by laryngoscopy prior to and after the operation. Basic patient characteristics, clinical parameters and follow-up data were reported and compared between groups defined by the different resection strategies. Histological entities were reported according to the 2017 World Health Organization (WHO) classification of neoplasms of the thyroid [14]. Surgical complications were reported according to the classification of surgical complications published by Dindo et al. [15]. Vocal cord paresis was defined as permanent if persisting for over 6 months. Kaplan–Meier survival analysis (for post-resection survival and recurrence-free survival) was performed using IBM SPSS Statistics 29.0.0.0 (IBM Corporation, Armonk, NY, USA). The log-rank (Mantel–Cox) test was used to analyze whether the mortality risk was different in the included patient groups. Statistical significance was assumed at *p* < 0.05 (two-sided). The tracheal resection strategies were classified as follows.

### 2.1. Non-Transmural Resection

#### Trachea Shaving

Trachea shaving is the most limited form of tracheal resection as it represents a superficial laminar ablation, without opening the tracheal lumen [13]. This technique was generally chosen in the case of superficial tumor infiltration.

### 2.2. Transmural Resection

Transmural tumor invasion required excision of the entire tumor-affected part of the tracheal wall. Depending on the extension of tumor infiltration, either partial “window” resection, near-circular resection, sleeve resection or total laryngectomy with extramucosal esophageal resection was performed.

#### 2.2.1. Partial “Window” and Near-Circular Resections

A partial “window” resection was performed if the tumor invasion involved less than one-fourth of the tracheal circumference and less than 2 cm in vertical dimension. Near-circular resections were performed if the major part of the tracheal circumference harbored tumor infiltration, which, yet, did not require sleeve resection. In the case of primary tracheal suturing of the defect, facilitated by tracheal mobilization and partial tracheal rotation, a reinforcement of the reconstruction was performed with a fascia-coated muscle rotation flap of strap muscles/the ipsilateral sternocleidomastoid muscle (medial and/or lateral caput) and/or a pectoralis major transposition flap. Thereby, tension was reduced, and potentially emerging minor tracheal leaks were primarily covered by the muscle flap(s) to reduce the need of an operative revision for emphysema and associated wound infection in the first place. Depending on the complexity of reconstruction, a protective tracheostomy was fashioned. In the case of the necessity to cover a remaining tracheal wall defect, a pectoralis major transposition flap of the ipsi- or contralateral side was utilized, with its adhering skin facing the tracheal lumen. For the latter, a protective tracheostomy was created in each case. Protective tracheostomies were maintained for at least 2 weeks.

#### 2.2.2. Sleeve Resection

Circumferential tracheal tumor invasions were treated by horizontal tracheal sleeve resection with tracheal mobilization and primary end-to-end anastomosis. Depending on the prevailing traction conditions, reinforcement of the reconstruction was carried out (by a transposition flap of the pectoralis major muscle) and protective tracheostomy was fashioned. Protective tracheostomies were maintained for at least 2 weeks.

#### 2.2.3. Total Laryngectomy with Extramucosal Esophageal Resection

In the case of total laryngectomy with extramucosal esophageal resection, a permanent tracheostomy was created. Total laryngectomy was required if bilateral laryngeal invasion and adjacent tracheal invasion > 5–6 cm vertically was present.

## 3. Results

The present cohort included 33 patients: 15 females and 18 males. The overall mean follow-up was 29.2 months (Table 1). The two-year postoperative survival rate of the overall cohort was 82.0 percent (Figure 2), while the two-year recurrence-free survival was 75.0 percent (Figure 3). The underlying histological entities were distributed as follows: 19 papillary thyroid carcinomas (PTC: 11 classical PTCs, 4 follicular variants of PTC, 2 mixed follicular/solid variants of PTC, 1 PTC with a partial component of poorly differentiated thyroid carcinoma and 1 PTC with a partial component of undifferentiated thyroid carcinoma), 8 follicular thyroid carcinomas (FTCs), 4 poorly differentiated thyroid carcinomas (PDTCs) and 2 oncocytic carcinomas (Table 1) [14].

The post-resection survival was most impaired in the case of PDTC, in comparison to oncocytic carcinoma and DTC, though not statistically significant (Figure 2). There were 20 R1 and 13 R0 resections. The two-year postoperative survival rate was 72.5 percent in the case of R1 resection, and 100.0 percent in the case of R0 resection; the difference was not statistically significant (Figure 2). R0 resection did not positively influence the recurrence-free survival in the present analysis (Figure 3).

There were 14 cases of trachea shaving and 19 cases of transmural trachea resection (9 “window” resections, 6 near-circular resections, 3 sleeve resections and 1 case of total laryngectomy with extramucosal esophageal resection). The post-resection survival was tendentially superior in the case of trachea sleeve resection (Figure 2). The two-year recurrence-free survival was higher in the case of trachea shaving and “window”/near-circular resection, compared to trachea sleeve resection (Figure 3).

### 3.1. Trachea Shaving

A total of 14 patients underwent trachea shaving. The sexes were equally distributed in this group (7:7, Table 1). Papillary thyroid carcinoma represented the primarily underlying histological entity in this group. In total, 10 of 14 patients underwent simultaneous laminar esophageal resection. To strengthen the tracheal wall after trachea shaving, a rotation flap of strap muscles/the sternocleidomastoid muscle was created in two patients (Table 2).

Resection of the recurrent laryngeal nerve was necessary in 11 of the 14 cases due to tumor infiltration. In these 11 cases, paresis of the recurrent laryngeal nerve was registered by preoperative laryngoscopy. A total of 12 patients had postoperative permanent vocal cord paresis (Table 3).

Within the group of trachea shaving, two tracheostomies were created: In one patient who underwent surgery with palliative intention (who suffered from infiltration of the larynx/esophagus and pulmonary metastases of a recurrent follicular thyroid carcinoma), a permanent tracheostomy was fashioned. The second patient underwent a tracheostomy due to bilateral signal reduction in intraoperative neuromonitoring after resection. The tracheostomy resulted as permanent because the patient died from multiorgan failure in the early postoperative period. One patient was reoperated on for a tracheal leak due to tracheal necrosis (Dindo Clavien grade 4 [15]), which required reconstruction with a sternocleidomastoid muscle flap during the reoperation.

### 3.2. Partial “Window” and Near-Circular Trachea Resections

Of the 19 patients who underwent transmural trachea resection, 15 patients were grouped for “window” (n = 9, Figure 4) and near-circular trachea resections (n = 6). Because the reconstruction technique for both “window” resection and near-circular resection was similar and regularly included mobilization, rotation of the trachea and muscle flap reinforcement, both resection types were analyzed in one group. The mean age was 62.9 years, and the sexes were similarly distributed (8:7, Table 1). The majority of patients in this group were operated on with curative intention (Table 1). The main histological entity in this group was papillary thyroid carcinoma. In total, 14 of 15 patients received a reinforcement of the tracheal reconstruction by using muscle flaps (mainly rotation flaps of strap muscles/sternocleidomastoid muscle, one pectoralis major transposition muscle flap and one latissimus dorsi transposition flap, Table 2). The mean intensive care unit stay was 9 days (Table 3). Two patients died in the early postoperative period (Dindo Clavien grade 5, Table 3). In one case, mechanical traction on the brachiocephalic trunk led to a fatal hemorrhage after interposition of the vessel between the reconstructed and thus shortened trachea and the posterior wall of the sternum. Henceforth, to avoid this complication, a pectoralis major flap was used in similar cases that allowed for an increase in the space between the trachea and the sternum to prevent vascular corrosion. In the second case, multiorgan failure was responsible for the patient’s death.

### 3.3. Sleeve Resection

Three patients (one female and two males) were treated by tracheal sleeve resection due to circumferential tumor infiltration (Table 1). In one case, an R0 resection was achieved, whereas in two cases, R1 resections resulted (Table 3). In all cases, temporary tracheostomies were created (Table 2). In one patient, muscle flap reinforcement was necessary. Only Dindo Clavien grade 1 complications were registered in this group (Table 3). The median intensive care stay was 14 days.

### 3.4. Total Laryngectomy

One patient underwent a total laryngectomy with extramucosal esophageal resection. Due to an infiltration of the larynx, the operation was initially planned as a laryngectomy with permanent tracheostomy, and an encompassing informed consent was obtained before the operation. At the time of the operation, the patient was 59 years old. The operation was performed with curative intention. The laryngectomy necessitated, in addition, a resection of the esophagus, leaving intact only the mucosa, which is the prerequisite for esophageal speech. The patient was observed at the intensive care unit for 2 days (Table 3).

## 4. Discussion

If thyroid carcinomas invade into the tracheal adventitia or the entire tracheal wall, compression, stenosis and tumor protrusion into the tracheal lumen can be the result [6]. Clinical symptoms may include dyspnea and hemoptysis [16]. An additional invasion of the infrahyoid muscles, recurrent laryngeal nerve, larynx, esophagus and/or vascular nerve sheath (vagal nerve, common carotid artery and internal jugular vein) can be observed. Historically seen, radical extirpation of thyroid carcinoma invading the trachea was associated with an extremely high morbidity and mortality, which is why—even nowadays—many abstain from performing extensive resections in the case of tumor invasion into the upper aerodigestive tract [17]. The treatment of these patients does not only require surgical expertise but also the background of a center provided with intensive care facilities, specialists in logopedic rehabilitation and nurses trained in taking care of patients with a tracheostomy and swallowing deficiency. Piazza et al. concluded from a systematic review of 656 tracheal and cricotracheal resections with end-to-end anastomosis for locally advanced thyroid cancer that these operations should be carried out by skilled surgical teams in tertiary referral centers, following critical patient selection, taking into account the operation-associated, potentially life-threatening risks [18].

In contrast to poorly differentiated and anaplastic thyroid carcinoma, well-differentiated thyroid carcinomas are less frequently diagnosed as the underlying histological entity in cases of locally advanced tumor stages (only in 5 to 13 percent) [10,19,20]. In the present analysis, the main histological entity was in fact differentiated thyroid carcinoma (81.8 percent, 27/33). One of the included PTC cases, however, harbored a PDTC component and another one a component of undifferentiated thyroid carcinoma, which might have negatively influenced the observed postoperative outcome in the DTC patient group of this study. Moreover, the WHO definition of thyroid neoplasms of 2017 was applied to characterize the underlying entities. A revision of the tumors according to the novel WHO definition of thyroid neoplasms of 2022 [21] might allow for the diagnosis of more aggressive differentiated high-grade thyroid carcinoma among these tumors, depending on the mitotic count and necrosis. In addition, the distribution of histological entities of the current analysis bears a selection bias due to the choice of external physicians to send patients with advanced thyroid carcinomas to the Section of Endocrine Surgery, UMC Mainz. Patients who underwent trachea resections for differentiated thyroid carcinoma had longer disease-specific survival when compared to the poorly differentiated type [13]. In the present analysis, the result was similar; post-resection survival was tendentially shorter for patients harboring poorly differentiated thyroid carcinoma. Wong et al. reported in 2019 that the extent of invasion is an important parameter that affects the clinical outcome for patients with PDTC: whereas patients with encapsulated PDTC with capsular invasion only or focal vascular invasion had an excellent outcome, the 5-year disease-free survival for patients with widely invasive PDTC was markedly reduced to only 10 percent [22]. Given the fact that patients with tracheal invasion represent highly advanced/invasive forms of PDTC, it appears sensible to critically call into question the indication to operate (or reoperate) in cases with the diagnosis of extensive PDTC being established before the intervention.

Different stages of tracheal invasion, which preoperatively can be described by results from panendoscopy, contrast-enhanced computed tomography and magnetic resonance imaging, require different surgical concepts, adapted to the extent of invasion into the aerodigestive tract [23]. In a study, which analyzed the outcome after trachea resection for advanced differentiated thyroid carcinoma, the simultaneous involvement of the trachea and the esophagus by differentiated thyroid carcinoma infiltration showed a higher tendency toward locoregional recurrence, whereas the involvement of the larynx was associated with lower disease-specific survival [24]. Laminar esophageal resections for superficial local tumor invasion were performed in 42.4 percent of the present cohort undergoing tracheal resections. Non-transmural resections (trachea shaving) are usually performed for superficial tumor invasion without opening the tracheal lumen and can be potentially complemented by extramucosal esophageal resections [13]. Tsukahara et al. observed overall 5- and 10-year survival rates of 93 percent and 41 percent in a cohort of 22 patients who underwent thyroid surgery with trachea shaving for papillary thyroid carcinoma [25]. In this analysis, trachea shaving was performed in 14 cases, and in 10 cases of these, additional esophageal resection was necessary. Larger transmural tumor invasion of the trachea requires an excision of the entire tracheal wall affected by the invasion. Brauckhoff et al. showed that tracheal window resection and sleeve resection yielded a longer disease-specific survival than non-transmural resections [13]. In this analysis, a tendency toward longer post-resection survival was visible for trachea sleeve resection. Yet, the two-year recurrence-free survival was superior in the case of trachea shaving and “window”/near-circular resection. Generally, due to the low number of patients included in this single-center series, the significance of the findings has to be interpreted critically. This observation is, however, in contrast to the previous findings that circular resections with end-to-end anastomosis were associated with longer recurrence-free survival when compared to “window resections” [17]. A meta-analysis by Allen et al. of 96 studies about tracheal resection for thyroid carcinoma, which analyzed the likelihood of recurrence in a detailed way, revealed that locoregional recurrence can be expected in 15 percent of patients after circumferential resection and in 25.6 percent of patients after window resection [16]. Distant recurrence, however, can be expected in 19.7 percent following circumferential resection and 15.6 percent after window resection [16]. Allen et al. concluded from this meta-analysis that neither the technique of sleeve resection with end-to-end anastomosis nor tracheal window resection can be recommended over the other [16]. In this meta-analysis, however, there is a potential bias due to the fact that the radicality of resection depends not only on the status of tumor infiltration but also on the decision of the operating surgeon and the willingness to accept the risk of a more advanced surgical approach.

In addition to the resection technique, the reconstruction and the reinforcement of the reconstruction by muscle flaps plays an important role: in the case of long-segment circular tracheal resections of more than 2 cm, tension-free anastomosis of the tracheal stumps is usually not possible [26]. Before performing the anastomosis, it is necessary to assess whether the tracheal stumps can be adapted without tension to prevent the sutures from tearing out and thereby damaging the tracheal stumps, which further enlarges the tracheal defect. Also, during the mobilization of the trachea, the blood supply of the trachea must be taken into account [17]. By pivoting parts of the trachea wall and/or longitudinal rotation of the tracheal ends, small gains in length can be achieved. This should already be considered when resecting the tumor and, if necessary, a step-shaped, spiral or oblique resection line should be chosen [26]. Asymmetrical resections are useful to preserve the contralateral recurrent laryngeal nerve in the case of resection in the laryngotracheal junction. In the case of infiltration of the ventral wall portions of the trachea, the pars membranacea can be preserved and used to reconstruct the wall defects [26]. To facilitate the tension-reduced healing of tracheal anastomoses, the external stitching of the skin between the chin and sternal manubrium in maximum flexion (sentinel stitch or “Grillo’s stitch”, named after the thoracic surgeon Hermes C. Grillo) was described as a useful addition during the initial postoperative interval [27,28,29]. However, we never used this technique because it was not necessary in the patients described. Moreover, it is extremely uncomfortable for the patients.

Due to radical tumor resections, the location of the common carotid artery and the brachiocephalic trunk result in the immediate vicinity of the anastomosis. Air leaks with the spread of bacteria into this area can subsequently lead to arrosion of the vessels’ walls, potentially causing fatal bleeding into the tracheobronchial system [30]. To prevent this lethal complication, it was recommended to interpose muscle flaps [26,31]. By covering a tracheal anastomosis or reconstruction with a well-vascularized muscle flap, the anastomosis can be reinforced, which may have a positive influence on the healing process [26]. It should be planned at the beginning of the operation which muscle flaps can be utilized for this purpose. First, the strap muscles can be used as a muscle flap [26]. The sternohyoid and sternothyroid muscles should be transected directly in the region of the sternum. Similarly, the sternocleidomastoid muscle can be used, either partially or completely [26]. For a partial muscle flap, the sternal attachment and the clavicular attachment of the muscle are split and transected at the attachment site. In the present cohort, a reinforcement of the anastomosis in the case of “window” and near-circular tracheal resections was performed in 93.3 percent, e.g., using muscle flaps of strap muscles or sternocleidomastoid muscle (11 of 14 cases). Moreover, the pectoralis major can be used as a pure muscle flap or myocutaneous flap. The key point is to preserve the vascular supply to the muscle. Central venous catheters should not be placed on the side of the muscle flap. A myocutaneous flap is suitable for the closure of large cervical skin defects. In the present cohort, this technique was used in two patients who underwent near-circular tracheal resection (in one case in combination with a local strap muscle flap) and in one patient who underwent tracheal sleeve resection. Furthermore, latissimus dorsi flaps can be used to cover large tissue defects. The muscle flap can be luxated to the cervical region via a subcutaneous tunnel, ventral to the clavicle [26]. This technique was used in one patient in this cohort who was treated with a “window” tracheal resection.

A study by Brauckhoff et al. demonstrated a statistically significant survival advantage when clear margins were achieved in the final pathology [13]. A study by Su et al., focused exclusively on differentiated thyroid carcinoma, did not find any statistically significant difference in overall survival or disease-free survival between resections with negative margins over resections with microscopically positive margins or gross disease [10]. In this study, R1 resections were performed in 60.6 percent of all cases (20/33). Recurrence-free survival was not influenced by the R status in the present analysis, whereas post-resection survival was tendentially longer in the case of R0 resections. A probable bias arises from the circumstance that the subgroup who received trachea shaving (i.e., the subgroup with generally less advanced carcinomas, compared to the patients who received transmural trachea resections) comprised 10 of the 20 R1 resections. To allow for major trachea reconstruction after transmural resections, clear margins should be aimed for, to allow for a proper healing of the defect, if this is feasible in the individual case. It has to be taken into consideration, though, that large resection zones complicate the reconstruction of the trachea in the final phase of the operation.

The complication rate of the present cohort was 45.5 percent. Of these, 27.3 percent were Dindo Clavien grade 1. The perioperative mortality within the present cohort, which also included operations with palliative intention, was 9.1 percent (3 of 33 patients). A systematic review by Piazza et al., who analyzed 656 patients who underwent tracheal and cricotracheal resection with end-to-end-anastomosis, revealed that a surgical complication rate of 27.7 percent can be expected, with a perioperative mortality of 2 percent [18]. In the present analysis, only three patients underwent tracheal sleeve resection. Among these patients, only complications of grade 1 (Dindo Clavien) were observed. Gaissert et al. observed a major reduction in the overall complication rate from 44 to 26 percent over three decades until 2005 following tracheal resections in a cohort of 82 patients suffering from advanced thyroid carcinoma [32].

Finally, in patients harboring distant metastatic disease (in the present cohort, 12.1 percent with distant metastases), radical or near-total resection of the invasive cervical cancer by resection and reconstruction of the upper airways may increase the quality of life and prevent death by suffocation.

## 5. Conclusions

Tracheal resections for locally advanced infiltration by thyroid carcinoma are feasible and represent an element of highly individualized treatment concepts. Patients need to be thoroughly informed about operation-associated risks and potentially life-threatening complications. By applying an individualized resection concept, adapted to the extent of tumor infiltration into the trachea and adjacent organs, suffocation due to an anticipated further tumor growth can be avoided, which may lead to prolonged survival. In patients with poorly differentiated histology, survival benefit following extensive trachea surgery remains, however, doubtful.

## Figures and Tables

**Figure 1 cancers-16-00163-f001:**
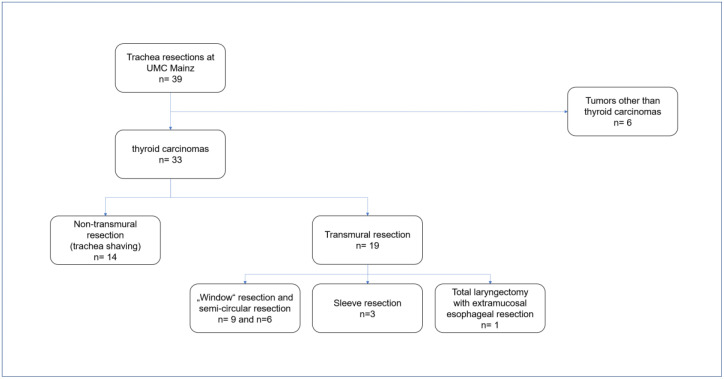
Flow chart—inclusion of patients with tracheal resections. Of 39 patients who underwent neck surgery including tracheal resections at the Section of Endocrine Surgery of the University Medical Center Mainz, 33 were treated for thyroid carcinoma. Other entities led to exclusion from the study.

**Figure 2 cancers-16-00163-f002:**
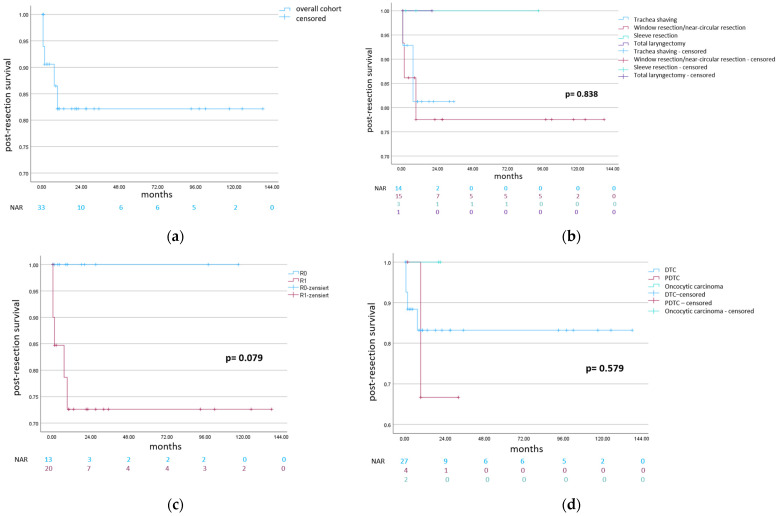
Kaplan–Meier analysis of post-resection survival: (**a**) post-resection survival of the overall cohort; (**b**) survival for the different types of surgical resection; (**c**) survival according to the resection margin (R0 versus R1); (**d**) survival according to the different histological entities (abbreviations: NAR = number at risk, DTC = differentiated thyroid carcinoma, and PDTC = poorly differentiated thyroid carcinoma).

**Figure 3 cancers-16-00163-f003:**
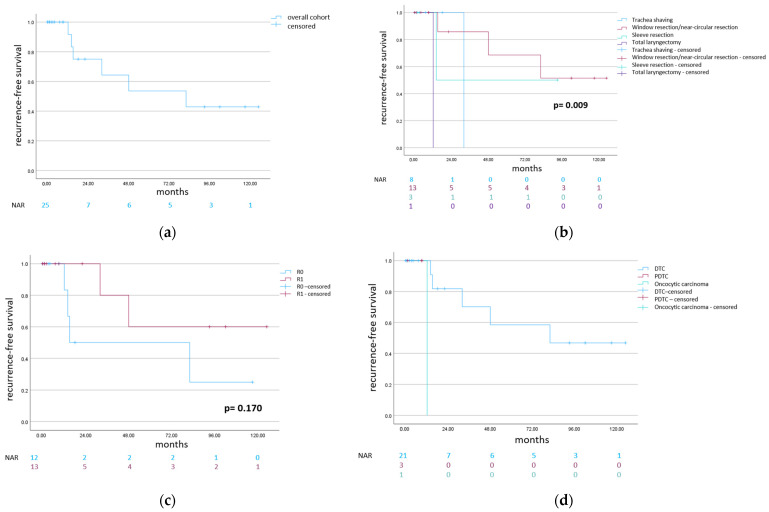
Kaplan–Meier analysis of recurrence-free survival. After exclusion of patients with postoperative tumor persistence, the cohort consisted of 25 patients. Deceased patients were regarded as censored. (**a**) Recurrence-free survival of the cohort; (**b**) recurrence-free survival for the different types of surgical resection. (**c**) shows the survival according to the resection margin (R0 versus R1). (**d**) indicates survival according to the different histological subtypes. Due to the low number of histological subtypes of PDTC and oncocytic carcinoma, log-rank test was not performed for 3d. (Abbreviations: NAR = number at risk, DTC = differentiated thyroid carcinoma, and PDTC = poorly differentiated thyroid carcinoma.)

**Figure 4 cancers-16-00163-f004:**
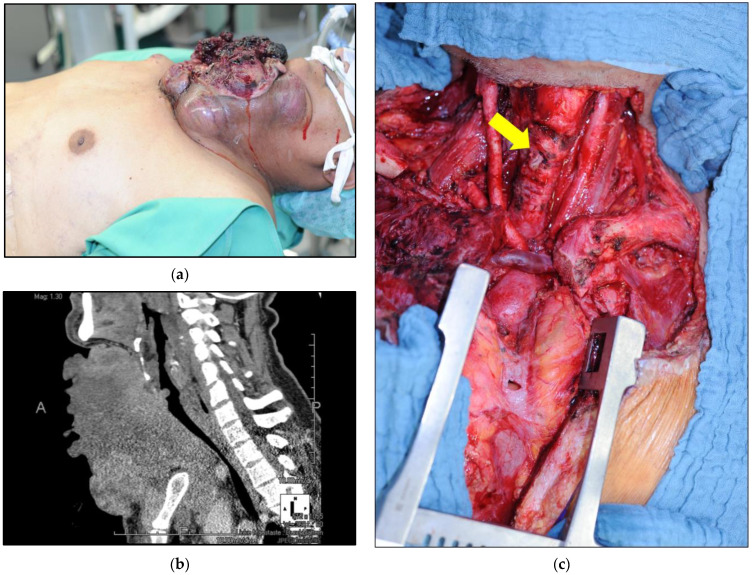
Tracheal “window” resection for papillary thyroid carcinoma—In a 54-year-old male patient with a large ulcerated papillary thyroid carcinoma. (**a**) Clinical image; (**b**) preoperative computed tomography scan; (**c**) a palliative tumor resection including thyroidectomy, lymphadenectomy, resection of the right internal jugular vein and right subclavian vein was performed. Due to tumor infiltration, the right sternoclavicular articulation was resected, too. A tracheal “window” resection was performed (yellow arrow). The tumor was classified as pT4b, pN1b (14/18), M1 (lung, bone), R1. The remaining defect was reconstructed using a latissimus dorsi muscle flap.

**Table 1 cancers-16-00163-t001:** Basic patient characteristics.

	Non-Transmural Resection	Transmural Resection			Total
	Trachea Shaving n = 14	“Window” Resection/ Near-Circular Resection n = 15	Sleeve Resection n = 3	Total Laryngectomy with Extramucosal Esophageal Resection n = 1	n = 33
Age (mean, range, years)	65.8 (23–88)	62.9 (17–85)	69.3 (60–78)	59.0 (59.0)	64.6 (17–88)
Sex ratio (M: F)	7:7	8:7	2:1	1:0	18:15
Previous thyroid operation history (n, %)	10 (71.4)	7 (46.7)	2 (66.7)	1 (100.0)	20 (60.6)
Preoperative recurrent laryngeal nerve paresis per patient (n, %)	11 (78.6)	8 (53.3)	1 (33.3)	1 (100)	21 (63.6)
Operative intention					
- Curative (n, %)	10 (71.4)	14 (93.3)	3 (100)	1 (100.0)	28 (84.8)
- Palliative (n, %)	4 (28.6)	1 (6.7)	0 (0)	0 (0)	5 (15.2)
Further treatment					
- Radioiodine (n, %)	6 (42.9)	9 (60.0)	2 (66.7)	1 (100.0)	18 (54.6)
- Radiation (n, %)	3 (21.4)	0 (0)	1 (33.3)	0 (0)	4 (12.1)
- Watch and wait (n, %)	5 (35.7)	6 (40.0)	0 (0)	0 (0)	11 (33.3)
Mean follow-up (months, range)	10.8 (0–35)	45.8 (0–138)	34.6 (2–93)	20 (20)	29.2 (0–138)
Histology [14]					
- Papillary thyroid carcinoma (n, %)	8 (57.1)	9 (60.0)	2 (33.3)	0 (0)	19 (57.6)
- Follicular thyroid carcinoma (n, %)	3 (21.4)	4 (26.7)	1 (33.3)	0 (0)	8 (24.2)
- Poorly differentiated thyroid carcinoma (n, %)	2 (14.3)	2 (13.3)	0 (0)	0 (0)	4 (12.1)
- Oncocytic carcinoma (n, %)	1 (7.1)	0 (0)	0 (0)	1 (100)	2 (6.1)
Tumor persistence after operation (n, %)	6 (42.9)	2 (13.3)	0 (0)	0 (0)	8 (24.2)
- Distant metastases (n, %)	2 (14.3)	2 (13.3)	0 (0)	0 (0)	4 (12.1)
- Local persistence (n, %)	3 (21.4)	0 (0)	0 (0)	0 (0)	3 (9.1)
- Distant and local persistence (n, %)	1 (7.1)	0 (0)	0 (0)	0 (0)	1 (3.0)
Recurrence	1 (7.1)	3 (20.0)	1 (33.3)	1 (100.0)	6 (18.2)
- Distant metastases (n, %)	0 (0)	1 (6.7)	1 (33.3)	1 (100.0)	3 (9.1)
- Local recurrence (n, %)	1 (7.1)	1 (6.7)	0 (0)	0 (0)	2 (6.1)
- Distant and local recurrence (n, %)	0 (0)	1 (6.7)	0 (0)	0 (0)	1 (3.0)

**Table 2 cancers-16-00163-t002:** Resection characteristics.

	Non-Transmural Resection	Transmural Resection			Total
	Trachea Shaving n = 14	“Window” Resection/ Near-Circular Resection n = 15	Sleeve Resection n = 3	Total Laryngectomy with Extramucosal Esophageal Resection n = 1	n = 33
Thyroid resection (n, %)	6 (42.9)	9 (60.0)	2 (66.7)	0 (0)	17 (51.5)
Lymphadenectomy					
- Compartment-oriented lymphadenectomy (n, %)	10 (71.4)	11 (73.3)	3 (100.0)	0 (0)	24 (72.7)
- Selective lymphadenectomy (n, %)	0 (0)	2 (13.3)	0 (0)	0 (0)	2 (6.1)
Esophagus resection (n, %)	10 (71.4)	3 (20.0)	0 (0)	1 (100.0)	14 (42.4)
Vascular resection (n, %)	4 (28.6)	1 (6.7)	0 (0)	0 (0)	5 (15.2)
Strap muscle resection (n, %)	10 (71.4)	5 (33.3)	2 (66.7)	0 (0)	17 (51.5)
Laryngeal muscle resection (n, %)	3 (21.5)	1 (6.7)	0 (0)	0 (0)	4 (12.2)
Laryngectomy					
- Total (n, %)	0 (0)	0 (0)	0 (0)	1 (100.0)	1 (3.0)
- Partial (n, %)	0 (0)	0 (0)	1 (33.3)	0 (0)	1 (3.0)
Resection of recurrent laryngeal nerve (n, %)	11 (78.6)	7 (46.7)	1 (33.3)	0 (0)	19 (57.6)
Thymectomy (n, %)	7 (50.0)	8 (53.3)	1 (33.3)	0 (0)	16 (48.5)
Protective tracheostomy (n, %)	0 (0)	8 (53.3)	3 (100.0)	0 (0)	11 (33.3)
Permanent tracheostomy (n, %)	2 (14.3)	4 (26.7)	0 (0)	1 (100.0)	7 (21.2)
Reconstruction with muscle flap (n, %)	2 (14.3)	14 (93.3)	1 (33.3)	0 (0)	17 (51.5)
- Rotation flap of strap muscles/sternocleidomastoid (n, %)	2 (14.3)	11 (73.3)	0 (0)	0 (0)	13 (39.4)
- Transposition flap of pectoralis major muscle (n, %)	0 (0)	1 (6.7)	1 (33.3)	0 (0)	2 (6.1)
- Combined pectoralis and strap muscle flap (n, %)	0 (0)	1 (6.7)	0 (0)	0 (0)	1 (3.0)
- Transposition flap of Latissimus dorsi muscle (n, %)	0 (0)	1 (6.7)	0 (0)	0 (0)	1 (3.0)

**Table 3 cancers-16-00163-t003:** Postoperative parameters and complication assessment.

	Non-Transmural Resection	Transmural Resection			Total
	Trachea Shaving n = 14	“Window” Resection/ Near-Circular Resection n = 15	Sleeve Resection n = 3	Total Laryngectomy with Extramucosal Esophageal Resection n = 1	n = 33
Median intensive care unit stay (days, range)	3 (0–30)	9 (0–35)	14(0–24)	2 (2)	7 (0–35)
Complications (n, %) [15]	7 (50.0)	5 (33.3)	2 (66.7)	1 (100.0)	15 (45.5)
- Dindo Clavien grade 1 (n, %)	4 (28.6)	2 (13.3)	2 (66.7)	1 (100.0)	9 (27.3)
- Dindo Clavien grade 2 (n, %)	1 (7.1)	1 (6.7)	0 (0)	0 (0)	2 (6.1)
- Dindo Clavien grade 3 (n, %)	0 (0)	0 (0)	0 (0)	0 (0)	0 (0)
- Dindo Clavien grade 4 (n, %)	1 (7.1)	0 (0)	0 (0)	0 (0)	1 (3.0)
- Dindo Clavien grade 5, death (n, %)	1 (7.1)	2 (13.3)	0 (0)	0 (0)	3 (9.1)
Registered time of death					
- Intraoperative period	0 (0)	0 (0)	0 (0)	0 (0)	0 (0)
- Perioperative period	1 (7.1)	2 (13.3)	0 (0)	0 (0)	3 (9.1)
- Postoperative period	0 (0)	0 (0)	0 (0)	0 (0)	0 (0)
Reason for death					
- Multiorgan failure	1 ^1^	1 ^2^	0	0	2
- Major bleeding of cervical vessels	0	1 ^3^	0	0	1
Median follow-up after surgery (months, range)	9 (0–35)	22 (0–138)	9 (2–93)	20 (20)	10 (0–138)
Resection margin					
- R0 (n, %)	4 (28.6)	7 (46.7)	1 (33.3)	1 (100)	13 (39.4)
- R1 (n, %)	10 (71.4)	8 (53.3)	2 (66.7)	0 (0)	20 (60.6)
Permanent vocal cord paresis per patient (n, % *)	12 (92.3)	8 (61.5)	1 (50.0)	1 (100.0)	22 (75.9)
Parathormone below 15 pg/mL in long-term follow-up (n, % **)	1 (7.7)	1 (6.7)	2 (66.7)	1 (100.0)	5 (16.7)

* Calculated for total patient number after exclusion of patients who died (3 patients with Dindo Clavien grade 5, 1 patient with trachea shaving and 2 with “window” resection) and who did not take part in postoperative laryngoscopy (1 patient with sleeve resection). ** calculated for total patient number after exclusion of patients without long-term analysis of parathormone. ^1^ One patient, who received an emergency operation including trachea shaving with palliative intention for a locally recurrent papillary thyroid carcinoma with preoperatively diagnosed metastases to the lung and bone and impeding suffocation due to local compression, died 31 days after the index operation from multiorgan failure, following bilateral malignant pleural effusion, pneumonia and respiratory and renal insufficiency. ^2^ One patient, who was treated with trachea window resection with palliative intention, died 48 days after the index operation from sepsis resulting from multi-drug-resistant bacteria that had colonized the large, ulcerated papillary thyroid carcinoma prior to surgery (confer Figure 4). ^3^ One patient, who underwent curative surgery including near-circular trachea resection for a papillary thyroid carcinoma with pulmonary metastases, died seven days after the index operation due to a major bleeding, due to arrosion of the brachiocephalic trunk against the sternum from the reconstructed and thus shortened trachea.

## Data Availability

The data presented in this study are available on request from the corresponding author. The data are not publicly available due to privacy restrictions.

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
