# Peer review of "A 16-Year Single-Center Series of Trachea Resections for Locally Advanced Thyroid Carcinoma"

_cancers, 2023, doi:10.3390/cancers16010163_

Round 1

Reviewer 1 Report

Comments and Suggestions for Authors

The study investigated surgical outcomes and clinical characteristics in patients with advanced thyroid carcinoma who underwent tracheal resections. The study included 33 patients (15 females and 18 males) who underwent neck surgery with trachea resections at the University Medical Center Mainz. The study included various histological entities of thyroid carcinoma, including papillary thyroid carcinomas, follicular thyroid carcinomas, poorly differentiated thyroid carcinomas, and oncocytic carcinomas.

Different tracheal resection strategies were used in the study, including trachea shaving, transmural trachea resections, such as "window" and near-circular resections, sleeve resections, and cervical evisceration. Post-resection survival varied among these strategies, with trachea sleeve resection showing tendentially superior outcomes.

Comments are listed as follows:

1.     Lack of Focus: The paper lacks a clear research focus or specific research questions. It is more of a descriptive study without a strong analytical component. A well-defined research question or objective is essential for scientific research.

2.     Insufficient Data Analysis: While the paper presents data, it lacks in-depth analysis and interpretation of the findings. Readers may find it challenging to understand the clinical implications and significance of the results without a more detailed analysis.

3.     Overly Detailed Sections: Some sections of the paper contain excessive detail about individual cases and complications. While clinical details are valuable, they should be presented in a more focused manner, and some of this detail might be better placed in a separate section or appendix.

4.     Inadequate Context: The paper lacks a sufficient introduction to the topic and fails to establish the context and importance of the research. It should provide a stronger rationale for the study.

5.     Statistical Significance: The paper mentions that some differences were not statistically significant, but it does not provide a comprehensive analysis of the statistical methods used or discuss the implications of these results.

6.     Clarity and Language: The paper's language could be improved for better readability and understanding. There are occasional inconsistencies in terminology, which should be addressed. An example is the word “framwork” should be “framework’.

7.     Citations and References: The paper needs to provide proper citations and references to support various claims and methodologies. For instance, some grading systems are mentioned without references. References were mostly published more than ten years ago and few in number.

Given these reasons, it might be beneficial to reject the paper in its current form and provide the authors with constructive feedback for improvement.

Comments on the Quality of English Language

Several typos were found scattered within the text. Nevertheless, the descriptions were clear and easy to follow.

Author Response

Reviewer #1

  1. Lack of Focus: The paper lacks a clear research focus or specific research questions. It is more of a descriptive study without a strong analytical component. A well-defined research question or objective is essential for scientific research.

Response: Even though the study is primarily descriptive, we do support the idea that it is helpful to publish the results of single centers, which perform these high-risk interventions in patients with thyroid carcinoma invading into the trachea. Other surgeons, who take into consideration to also perform this kind of surgery for advanced thyroid carcinomas but do not have a large experience, may use publications as e.g. the present one, as a guideline. Since this kind of operation is very rarely performed (partly due the associated consequences, feared by the operating team, partly due to the prevalence of such advanced thyroid carcinomas), a description of potential surgical treatment options can, in our opinion, add important information, which we discussed with regard to other, similarly concise descriptive studies from other centers. Furthermore, the reviewer asked for a clear focus, which is why we stated the research question in the “simple summary” in a more precise manner:

“Infiltration of the aerodigestive tract determines the prognosis in advanced thyroid carcinoma, ultimately leading to suffocation. To avoid this fatal outcome, surgical treatment strategies – within the framework of multidisciplinary patient care - are required. Since high-stage thyroid carcinoma with trachea infiltration is rare, there are no official guidelines to rely on, but individualized surgical treatment concepts are required. The aim of the study was to present and critically analyze the stepwise escalated surgical treatment strategy, which was applied in a 16-year time period at the University Medical Center Mainz, and the resulting post-resection and recurrence-free survival.”

  1. Insufficient Data Analysis: While the paper presents data, it lacks in-depth analysis and interpretation of the findings. Readers may find it challenging to understand the clinical implications and significance of the results without a more detailed analysis.

Response: We are very sorry that the reviewer does not consider or analysis an in-depth analysis. We described the patient cohort and operation technique in detail, analyzed post-resection survival and recurrence-free survival, reported operation-associated complications and critically described the patients, who unfortunately deceased perioperatively. The results are broadly discussed with regard to similar studies. During the revision, the discussion was expanded and deepened (confer changes marked in yellow). Currently, we cannot think of other details, which we did not closely describe or discuss in association with this type of surgery. If the reviewer asks for specific details that we have not yet checked or discussed, he or she can clarify this and we will in any case try to address the issue.

  1. Overly Detailed Sections: Some sections of the paper contain excessive detail about individual cases and complications. While clinical details are valuable, they should be presented in a more focused manner, and some of this detail might be better placed in a separate section or appendix.

Response: We have improved the structure of the manuscript, e.g. by introducing an addition to Table 3, in which the time point and the underlying reasons for the deceased patients’ deaths are described. This was also a suggestion by the Guest Editors. Generally, we believe that the critical reporting of complications is necessary in this field of high-risk, individualized surgery. Since the study gives an overview over cases, which were, one by one, operated with a tailored, patient- and disease-adapted operation technique, the reporting of details is an important aspect of this work. It should become clear, that there is not “one” operation method to treat advanced thyroid carcinomas with invasion into the trachea. In this sense, we would suggest to keep the details in the manuscript.

  1. Inadequate Context: The paper lacks a sufficient introduction to the topic and fails to establish the context and importance of the research. It should provide a stronger rationale for the study.

Response: Since trachea resections are rare (partly due the associated consequences, feared by the operating team, partly due to the prevalence of such advanced thyroid carcinomas), we believe that the publication of single center experience is valuable. In comparison to other studies, describing this kind of high-risk surgery for advanced thyroid carcinoma, the patient count of our present study is comparable and not too low, to be representative. Furthermore, point 4 brought up by the reviewer is similar to the point 1. Addressing this, we stated the research question in the “simple summary” in a more precise manner:

“Infiltration of the aerodigestive tract determines the prognosis in advanced thyroid carcinoma, ultimately leading to suffocation. To avoid this fatal outcome, surgical treatment strategies – within the framework of multidisciplinary patient care - are required. Since high-stage thyroid carcinoma with trachea infiltration is rare, there are no official guidelines to rely on, but individualized surgical treatment concepts are required. The aim of the study was to present and critically analyze the stepwise escalated surgical treatment strategy, which was applied in a 16-year time period at the University Medical Center Mainz, and the resulting post-resection and recurrence-free survival.”

  1. Statistical Significance: The paper mentions that some differences were not statistically significant, but it does not provide a comprehensive analysis of the statistical methods used or discuss the implications of these results.

Response: The term “statistically significant” or “not significant” is used with reference to the Kaplan Meier Survival analysis and the log-rank (Cox-Mantel) test, which was used to compare the analyzed groups included in each of the presented Kaplan Meier curves. We have now clarified and described the statistical analysis in detail in the “Methods” section. We also clearly stated in the “Discussion” section that - due to the low number of patients included in the study - the significance has to be interpreted critically. However, we pointed out that tendencies can be deduced from this analysis, which may be helpful to decide about potential trachea resections in specific patient cases.

  1. Clarity and Language: The paper's language could be improved for better readability and understanding. There are occasional inconsistencies in terminology, which should be addressed. An example is the word “framwork” should be “framework’.

Response: We have critically revised and corrected the spelling mistakes throughout the manuscript.

  1. Citations and References: The paper needs to provide proper citations and references to support various claims and methodologies. For instance, some grading systems are mentioned without references. References were mostly published more than ten years ago and few in number.

Response: The rarity of comparable research articles illustrates the importance of a publication of the results from our single center. Missing quotations of grading systems (e.g. Dindo Clavien Classification, UICC Classification, WHO classification of neoplasms of the thyroid) were now added.

Reviewer 2 Report

Comments and Suggestions for Authors

good work. The manusript is well written and the purpose is well defined and explained. please in the introduction when you report this sentence "... followed by the larynx and the esophagus [2-5]", you should specify the percentage of these kinds of cancers. About this you could mention this manuscript:

"Sireci F, Lorusso F, Dispenza F, Immordino A, Gallina S, Salvago P, Martines F, Bonaventura G, Uzzo ML, Spatola GF. A Prospective Observational Study on the Role of Immunohistochemical Expression of Orphanin in Laryngeal Squamous Cell Carcinoma Recurrence. J Pers Med. 2023 Jul 30;13(8):1211."

Author Response

Reviewer#2

good work. The manuscript is well written and the purpose is well defined and explained. please in the introduction when you report this sentence "... followed by the larynx and the esophagus [2-5]", you should specify the percentage of these kinds of cancers. About this you could mention this manuscript:

"Sireci F, Lorusso F, Dispenza F, Immordino A, Gallina S, Salvago P, Martines F, Bonaventura G, Uzzo ML, Spatola GF. A Prospective Observational Study on the Role of Immunohistochemical Expression of Orphanin in Laryngeal Squamous Cell Carcinoma Recurrence. J Pers Med. 2023 Jul 30;13(8):1211."

Response: We included the reference, as suggested by the reviewer, to underline that the vast majority of larynx-infiltrating cancers are squamous cell carcinomas. We specified the percentage of observed tumor infiltration into the abovementioned structures, originating from thyroid carcinoma, the entity, which is analyzed exclusively in this study:

“For invasive thyroid carcinoma, the trachea was described as the most common site of tumor invasion (incidence of 35–60 percent among the patients with invasive tumor growth), followed by invasion into the esophagus (21 percent) and the larynx (12 percent) [4-8]. In comparison, 85–95% of cancers affecting the larynx originate from the entity of head and neck squamous cell carcinoma [9].”

Reviewer 3 Report

Comments and Suggestions for Authors

The subject is quite important for endocrine surgeons, especially, because these cases are quite rare and it is very difficult to develop a personal experience for these very difficult cases. I suggest to you few correction in some parts of the article.

Peer review for paper “A 16-year single-center series of trachea resections for locally advanced thyroid carcinoma

In this paper, the authors present the resection strategies and clinical outcome of patients undergoing surgery for advanced thyroid carcinoma with trachea invasion. The study was realized at the single-center of the University Medical Center Mainz, in a 16-year time period. The subject is quite important for endocrine surgeons, especially, because these cases are quite rare and it is very difficult to develop a personal experience for these very difficult cases.

1. Title – The title is suggestive enough for the topic of the article.

2. The abstract is informative enough.

3. Keywords –adequate.

4. The introduction presents the known data on this subject and the reason for this study. Some information are not totally correct:

- row 35-36: in stage I thyroid cancer – ”generally allows for curation after thyroidectomy (with central lymphadenectomy)” – central lymphadenectomy is not necessary and performed in all the cases

5. Material and method: The study design- it should be mentioned that this is a retrospective study.

- row 75 – ”Permanent hypoparathyroidism was defined as parathormone level <15 pg/ml in longterm follow-up” -  correct low PTH and hypocalcemia

- 2.3. Partial “window” and near-circular resections, 2.4. Sleeve resection and 2.5. Cervical evisceration are parts of 2.2. Transmural resection, the sub-chapter number should be correct.

6. Results – The results obtained from 33 patients operated for advanced thyroid cancer are presented. The table 1 depicted the baseline data of the two cohort. The abreviations : PTC, FTC and PDTC should be explained under the table.

Figure 2 with Kaplan-Meier analysis of post-resection survival is not well inserted, so part of two graphs is out of the page.

The table 2 present in details resection characteristics.

The table 3 present the postoperative complications. I suggest that the Clavien-Dindo classification of surgical complications should be described in the text, maybe in the Method part.

The paragraph ”There were 14 cases of trachea shaving, 19 cases of transmural trachea resection (9 “window” resections and 6 near-circular resections (comprised in one group), 3 sleeve resections and 1 case of cervical evisceration).” is not very clear, there are too many parenthesis.

In row 152: ”Papillary thyroid microcarcinomas represented the underlying histological entity in this group” is not correct for patients operated by trachea shaving. It should be corrected.

7. The discussions address the subject of the paper, but the limits of this study should be mentioned. It is a retrospective study and the number of the cases is low. The value of this analysis should be underline, too.

8. Conclusions should be reformulated:

”Tracheal resections for locally advanced tumor infiltration” – thyroid is missing, it is about thyroid cancer

”By applying an individualized resection concept, adapted to the extent of tumor infiltration into the trachea and adjacent organs, suffocation due to further tumor growth can be avoided, thus ensuring prolonged survival. – the last part is not indicated by the results of this study, I suggest to reformulate.

9. The references are written according to the instructions for authors.

10. The English language is good, in my opinion, but I am not a native English speaker. May be in some parts of this article, the understanding of the text is quite difficult.

At the end, I wonder how Informed consent was obtained from all subjects involved in the study, if the study is retrospective and some patients died.

Author Response

Reviewer #3

The subject is quite important for endocrine surgeons, especially, because these cases are quite rare and it is very difficult to develop a personal experience for these very difficult cases. I suggest to you few corrections in some parts of the article.

  1. Title– The title is suggestive enough for the topic of the article.
  2. The abstractis informative enough. 
  3. Keywords–adequate 
  4. The introductionpresents the known data on this subject and the reason for this study. Some information are not totally correct:

- row 35-36: in stage I thyroid cancer – ”generally allows for curation after thyroidectomy (with central lymphadenectomy)” – central lymphadenectomy is not necessary and performed in all the cases

Response: We clarified in the cited sentence:

 “which generally allows for curation after thyroidectomy (potentially complemented by central lymphadenectomy) and radioiodine therapy in case of differentiated thyroid carcinoma.”

  1. Material and method: The study design- it should be mentioned that this is a retrospective study.

Response: We added the information in the initial sentence of the “Material and Methods” section:

“From 2007 to 2023, all patients undergoing neck surgery including trachea resections at the section of endocrine surgery of the single-center UMC Mainz were analyzed in a retrospective study.”

- row 75 – ”Permanent hypoparathyroidism was defined as parathormone level <15 pg/ml in longterm follow-up” -  correct low PTH and hypocalcemia

Response: As explained by the Guest Editors, hypoparathyroidism should be defined by the intake of calcium and vitamin D supplements. Unfortunately, we did not register this, which is why the definition of hypoparathyroidism was removed from the “Material and Methods” section. In Table 3, however, we kept the information about PTH values below the normal range in long-term follow-up, but without weighing this explicitly as “hypoparathyroidism”.

- 2.3. Partial “window” and near-circular resections, 2.4. Sleeve resection and 2.5. Cervical evisceration are parts of 2.2. Transmural resection, the sub-chapter number should be correct.

Response: We corrected the sub-chapters of transmural resections and thank the referee for informing us about this mistake.

  1. Results– The results obtained from 33 patients operated for advanced thyroid cancer are presented. The table 1 depicted the baseline data of the two cohort. The abreviations : PTC, FTC and PDTC should be explained under the table.

Response: We replaced the abbreviations by the original terms “papillary thyroid carcinoma, “follicular thyroid carcinoma” and “poorly differentiated thyroid carcinoma”.

Figure 2 with Kaplan-Meier analysis of post-resection survival is not well inserted, so part of two graphs is out of the page.

Response: We corrected the layout.

The table 2 present in details resection characteristics.

The table 3 present the postoperative complications. I suggest that the Clavien-Dindo classification of surgical complications should be described in the text, maybe in the Method part.

Response: We added the following explanatory sentence in the “Methods” section:

“Surgical complications were reported according to the classification of surgical complications published by Dindo et al. [13].“

The paragraph ”There were 14 cases of trachea shaving, 19 cases of transmural trachea resection (9 “window” resections and 6 near-circular resections (comprised in one group), 3 sleeve resections and 1 case of cervical evisceration).” is not very clear, there are too many parenthesis.

Response: We structured the sentence in the following way, taking also the Editors suggested terminology into account:

“There were 14 cases of trachea shaving and 19 cases of transmural trachea resection (9 “window” resections, 6 near-circular resections, 3 sleeve resections and 1 case of total laryngectomy (with extramucosal esophageal resection)).”

In row 152: ”Papillary thyroid microcarcinomas represented the underlying histological entity in this group” is not correct for patients operated by trachea shaving. It should be corrected.

Response: We corrected the sentence as follows:

“Papillary thyroid carcinoma represented the primarily underlying histological entity in this group.”

  1. The discussionsaddress the subject of the paper, but the limits of this study should be mentioned. It is a retrospective study and the number of the cases is low. The value of this analysis should be underline, too.

Response: We pointed out the limits of the study and clearly stated in the “Discussion” section that - due to the low number of patients included in the study - the results have to be interpreted critically. Furthermore, we pointed out that the paucity of data may lead to a bias, e.g. concerning the results of the impact of the R-status:

“In this study, R1 resections were performed in 60.6 percent of all cases (20/33). Recurrence-free survival was not influenced by the R status in the present analysis, whereas post-resection survival was tendentially longer in case of R0 resections. A probable bias arises from the circumstance that the subgroup who received trachea shaving (i.e. the subgroup with generally less advanced carcinomas, compared to the patients who received transmural trachea resections) comprised 10 of the 20 R1 resections.”

  1. Conclusionsshould be reformulated:

”Tracheal resections for locally advanced tumor infiltration” – thyroid is missing, it is about thyroid cancer

 ”By applying an individualized resection concept, adapted to the extent of tumor infiltration into the trachea and adjacent organs, suffocation due to further tumor growth can be avoided, thus ensuring prolonged survival. – the last part is not indicated by the results of this study, I suggest to reformulate.

Response: We reformulated the inaccuracies, which were pointed out by the reviewer:

“Tracheal resections for locally advanced infiltration by thyroid carcinoma are feasible and represent an element of highly individualized treatment concepts.”

and

“By applying an individualized resection concept, adapted to the extent of tumor infiltration into the trachea and adjacent organs, suffocation due to an anticipated further tumor growth can be avoided, which may lead to prolonged survival.”

  1. The references are written according to the instructions for authors.

  1. The English language is good, in my opinion, but I am not a native English speaker. May be in some parts of this article, the understanding of the text is quite difficult.

At the end, I wonder how Informed consent was obtained from all subjects involved in the study, if the study is retrospective and some patients died.

Response: Patients undergoing treatment at our center sign a declaration of consent in which they agree to the use of clinical data for institutional analyses. This is done prior to admission for surgery.

Reviewer 4 Report

Comments and Suggestions for Authors

In a retrospective observational study spanning from 2007 to 2023, researchers investigated the impact of tracheal resections on patients with advanced thyroid carcinomas at UMC Mainz, Germany. The study included 33 patients who underwent neck surgery with trachea resections for locally advanced thyroid carcinomas. The surgical procedures varied, with 14 patients undergoing non-transmural (trachea shaving) and 19 undergoing transmural trachea resections. The latter included "window" resections, near-circular resections, sleeve resections, and one cervical evisceration. The two-year postoperative survival rate was 82.0 percent, and the two-year recurrence-free survival rate was 75.0 percent, with a mean follow-up period of 29.2 months. The findings suggest that tracheal resections can be considered a feasible component of highly individualized treatment concepts for locally advanced tumor infiltration, emphasizing the importance of tailoring surgical approaches based on the specific stage of tracheal tumor invasion.

The manuscript is skillfully composed and succinct, addressing a crucial clinical topic with limited existing studies, thereby bridging a significant gap in the literature. I do however have a few remarks for the authors to consider:

1. "In 2018, the incidence of thyroid carcinoma in Germany was 4270 for females and 1930 for males". Incidence is usually a rate. Or do the authors refer to prevalence, I assume?

2. The authors rely on UICC staging, so I suggest that the AJCC/UICC version used to stage thyroid cancer is referenced. Which version (7th? 8th?) There are some differences regarding extrathyroidal extension et.c. that may need to be discussed.

3. In this study, one case of papillary thyroid carcinoma (PTC) exhibited a focal poorly differentiated thyroid carcinoma (PDTC) component, while a separate PTC showcased a regional anaplastic component. Consequently, I contend that these two samples should not be grouped together with other PTCs. Instead, they should be analyzed independently. Alternatively, it is advisable to address this distinction in the discussion, emphasizing that the PTC category encompasses a diverse range of entities.

4. The main histological entity in this study was differentiated thyroid carcinoma (81.8 percent, 27/33). I acknowledge that the 2022 WHO classification was not used to diagnose these lesions, but I still think that subsets of the PTCs/FTCs with tracheal invasion in this study would be reclassified into differentiated high-grade thyroid carcinoma if mitotic counts and necrosis were accounted for. As the main scope of this work is surgical, it would be to much of a maneuver to ask the authors to include this information, but please bring it up in the Discussion section - as histology can pinpoint poor-prognosis cases better with the 2022 WHO criteria than the 2004 or 2017 criteria. Just by looking at the macroscopic tumor in Figure 4, this lesion appears to be high-grade.

Author Response

Reviewer #4

The manuscript is skillfully composed and succinct, addressing a crucial clinical topic with limited existing studies, thereby bridging a significant gap in the literature. I do however have a few remarks for the authors to consider:

  1. "In 2018, the incidence of thyroid carcinoma in Germany was 4270 for females and 1930 for males". Incidence is usually a rate. Or do the authors refer to prevalence, I assume?

Response: We actually reported the raw incidence registered and published by the Robert Koch Institute (RKI) in Germany for the year 2018. Since this way of reporting the incidence is uncommon, as pointed out by the reviewer, we now report the incidence per 100 000, which was published by the RKI in the same online document:

“In 2018, the incidence of thyroid carcinoma in Germany was 10.2 per 100 000 females and 4.7 per 100 000 males [1].”

  1. The authors rely on UICC staging, so I suggest that the AJCC/UICC version used to stage thyroid cancer is referenced. Which version (7th? 8th?) There are some differences regarding extrathyroidal extension et.c. that may need to be discussed.

Response: The Robert Koch Institute (RKI), as a German federal government agency/ research institute responsible for disease control and prevention, appeared to us as a useful source to get an initial overview over the incidence and prevalence of thyroid carcinoma in Germany. The quoted data from the Robert Koch Institute concerning the incidence rate refer to the 8th edition of the UICC staging system. The data from the Robert Koch Institute reporting to the 5-year survival are, however, based on the 7th edition. We now correctly referenced the different UICC editions in the passage of the “Introduction” section.

  1. In this study, one case of papillary thyroid carcinoma (PTC) exhibited a focal poorly differentiated thyroid carcinoma (PDTC) component, while a separate PTC showcased a regional anaplastic component. Consequently, I contend that these two samples should not be grouped together with other PTCs. Instead, they should be analyzed independently. Alternatively, it is advisable to address this distinction in the discussion, emphasizing that the PTC category encompasses a diverse range of entities.

Response: We addressed this issue in the discussion section:

“In the present analysis, the main histological entity was in fact differentiated thyroid carcinoma (81.8 percent, 27/33). One of the included PTC cases, however, harbored a PDTC component and another one a component of undifferentiated thyroid carcinoma, which might have negatively influenced the observed postoperative outcome in the DTC patient group of this study.”

  1. The main histological entity in this study was differentiated thyroid carcinoma (81.8 percent, 27/33). I acknowledge that the 2022 WHO classification was not used to diagnose these lesions, but I still think that subsets of the PTCs/FTCs with tracheal invasion in this study would be reclassified into differentiated high-grade thyroid carcinoma if mitotic counts and necrosis were accounted for. As the main scope of this work is surgical, it would be too much of a maneuver to ask the authors to include this information, but please bring it up in the Discussion section - as histology can pinpoint poor-prognosis cases better with the 2022 WHO criteria than the 2004 or 2017 criteria. Just by looking at the macroscopic tumor in Figure 4, this lesion appears to be high-grade.

Response: We addressed this issue in the discussion section:

“(…) the WHO definition of thyroid neoplasms of 2017 was applied to characterize the underlying entities. A revision of the tumors according to the novel WHO definition of thyroid neoplasms of 2022 [21] might allow for the diagnosis of more aggressive differentiated high-grade thyroid carcinoma among these tumors, depending on mitotic count and necrosis.”

Reviewer 5 Report

Comments and Suggestions for Authors

Comments:

1. The study's sample size is insufficient. Kindly elaborate on the implications of this limited sample size in understanding its impact on the field of cancer research.

2. Kindly provide additional comprehensive information regarding patient characteristics. Such as BIO, smoking and alcohol history, etc.

3. Do all patients originate from Germany? If so, is there a particular region within Germany that they come from? Possible to get more patients' info from EU?

Author Response

Reviewer #5

  1. The study's sample size is insufficient. Kindly elaborate on the implications of this limited sample size in understanding its impact on the field of cancer research.

Response: Since trachea resections are rare (partly due the associated consequences, feared by the operating team, partly due to the prevalence of such advanced thyroid carcinomas), we believe that the publication of single center experience is valuable. In comparison to other studies, describing this kind of high-risk surgery for advanced thyroid carcinoma, the patient count of our present study appears comparable and definitely not too low, to be representative. Yet, we clarified in the “discussion” section, that the statistical significance of the results is limited and has to be interpreted critically, due to the relatively low number of patients included.

  1. Kindly provide additional comprehensive information regarding patient characteristics. Such as BIO, smoking and alcohol history, etc.

Response: We did not register smoking or alcohol history. To our knowledge, especially smoking and alcohol history can be associated with the development of SCC of airway/esophagus, but not with thyroid carcinoma invading into trachea or esophagus, which is why we did not register this information.

  1. Do all patients originate from Germany? If so, is there a particular region within Germany that they come from? Possible to get more patients' info from EU?

Response: The patients operated on were from different regions in Germany, but partly also from other countries. Therefore, we cannot draw conclusions from the geographical background. Furthermore, our intention was to analyze the surgical treatment of thyroid carcinoma of thyroid with invasion into the airway. The study, due to the restricted patient number, is not qualified to allow for broader epidemiological/geographical conclusions.

Round 2

Reviewer 1 Report

Comments and Suggestions for Authors

I have completed the thorough review of the manuscript titled "A 16-year single-center series of trachea resections for locally advanced thyroid carcinoma" (Manuscript ID: cancers-2692394) submitted to Cancers. I want to commend the authors for their diligence in addressing the concerns raised during the review process.

After careful consideration of the revised manuscript and the authors' responses to the critiques, I am pleased to convey my approval for the acceptance of the manuscript for publication in Cancers. The authors have successfully incorporated constructive feedback, enhancing the overall quality and clarity of their work.

Specifically, the revisions made in response to the initial concerns have significantly strengthened the manuscript in terms of focus, data analysis, detailed sections, context establishment, statistical significance, clarity of language, and proper citations and references. The refined research question, improved structure, and detailed analysis contribute to the scientific merit of the study.

I appreciate the authors' responsiveness to the feedback provided, and I believe that the manuscript is now suitable for publication in Cancers. I would like to express my gratitude to the authors for their dedication to refining and improving their work.

Author Response

Thank you very much for your positive report. 

Best regards

Reviewer 5 Report

Comments and Suggestions for Authors

No more comments

Author Response

First, we would like to thank the reviewer for the positive feedback. We hope that with the detailed responses to the editors requested minor revisions and associated changes in the manuscript, the reviewer is overall satisfied with the article in its current form. 

Best regards